# Foster Grandparent Programs’ Impact on the Quality-of-Life of Older Adult Volunteers

**DOI:** 10.3390/healthcare13030230

**Published:** 2025-01-24

**Authors:** Anastacia Schulhoff, Alex Dukehart

**Affiliations:** Department of Sociology, Appalachian State University, Boone, NC 28608, USA; alexdukehart@icloud.com

**Keywords:** quality of life (QoL), community participation, volunteering, older adults, United States, Foster Grandparent Program

## Abstract

**Background/Objectives:** Volunteering among older adults in the United States is rising, with rates up by 65% since 1974. With this tremendous growth in volunteering among seniors, examining how it affects them is essential. One such volunteering program we look at in this research is the Foster Grandparents Program (FGP) of the High Country, which allows older adults to volunteer at local schools to help in-need children. Previous research has shown how volunteering as an older adult positively impacts people’s quality of life. This paper will explore the quality-of-life changes for the FGP older adult volunteers, explicitly comparing the quality-of-life changes with demographics and program satisfaction. **Methods:** This study examined the survey response of 93 participants in the FGP. Cross-tabulation and correlations evaluated the relationships between demographic variables, satisfaction with the volunteer experience, and overall changes in the quality of life of volunteers. **Results:** Age, gender, and years of service were not good predictors of quality-of-life changes. Satisfaction with the FGP program proved to be the best predictor for change in quality of life. **Conclusions:** Volunteer program management has positive effects on volunteers self-reported quality-of-life changes. Additional studies could look at how this impacts the volunteers’ sense of purpose and its impact upon the time they donate to the volunteer program.

## 1. Introduction

Volunteering has a positive impact on the socialization and health of seniors. Volunteering has also been found to increase self-esteem and improve volunteers’ satisfaction with life, thus also impacting their sense of well-being [1]. Last year the Senior Corps, an organization working with seniors to help them find ways of volunteering, had over 200,000 senior volunteers across the nation [2]. Since 1974, volunteer rates among seniors, adults 55 and older, has gone up by 65% and these volunteers tend to commit twice the volunteer hours compared to younger volunteers [3]. For decades now, studies have shown that volunteering has a positive impact on older adult’s sense of well-being, quality of life and physical health [4,5,6]. It is, therefore, beneficial to provide supportive opportunities to older adults to volunteer, as people are living longer and healthier lives.

The Foster Grandparents Program (FGP) is one of the programs ran by the Appalachian Senior Programs and funded by Senior Corps. The FGP allows older adults to provide support services to youth who have special emotional, social, or educational needs. Older adult volunteers in these programs devote 20 to 40 h of their week to help these children. The Appalachian Senior Programs is federally funded by the Senior Corps program and allows older adults within the High Country consisting of Allegheny, Ashe, Avery, Watauga and Wilkes counties of North Carolina (N.C.) the opportunity to volunteer time to help in their community. The Appalachian Foster Grandparents Program (FGP) has been helping older adults volunteer in the High Country for over 30 years, and this program significantly influences the community, but more importantly, it impacts the older volunteers who serve the community. This N.C. community-based program impacts about 100 older adult volunteers and helps about 428 children [7]. Volunteering through this program is important, as it betters the lives of the volunteers and the “at-risk” children they assist.

The benefits are reciprocal to both the volunteers and the children that they assist in the K-12 school system. FGP engages people 55 and above and can help them stay active and enrich their lives. Volunteers receive benefits such as a tax-exempt hourly stipend, mileage reimbursements, and paid vacation and sick leave. They also receive accident and personal liability insurance. The FGP also provides one-on-one support to children with special needs, which can help improve their academic, social, or emotional development. Volunteers can help children learn to read, individually tutor them, mentor teenagers, care for premature infants or children with disabilities, and help children who have been abused or neglected.

### 1.1. Impact of Volunteerism

Volunteering has been shown to have a positive psychosocial impact on one’s sense of well-being, happiness in life, and even their health [7]. The act of volunteering has been shown to have many positive effects on the elderly, but there are still more questions to be asked. According to researchers who apply Activity Theory to their research on older adult volunteers, it is shown to have great benefits to older adults because it bridges the gap between working and retirement [8,9]. Other researchers show that by having new motivations for seniors, they can enhance their self-identity by moving into the role of “senior volunteer” or “foster grandparent” [10].

Volunteerism has been a focus in the field of gerontology for years. There have been many findings on the benefits of volunteering for the volunteers themselves [11]. These benefits affect a large portion of older adults and impact seniors’ health and well-being. These results have been attributed to many sociological theories, including Activity, Self-Determination, and Socioemotional Selectivity theories. In recent years, volunteerism has seen many changes in who volunteers and the number of hours each volunteer gives.

Trends show that those who have demonstrated a practice of volunteering throughout their life do not slow down in older age as paid work declines [12]. This trend continues in new volunteers, rating volunteer compensation as a low factor in joining [13]. While younger adult volunteer participation is decreasing, the opposite is happening with older adults. Volunteering in older adults is up 64% from 1976, with 23.5% of those older than 65 volunteering [8,14]. With the baby boomer generation retiring, this is expanding the volunteer pool, and many believe they will reinvent volunteering as organizations work with this new, larger, diverse group [15]. Within the baby boomer generation there is not one age group that can be predicted to volunteer more than another [16,17]. While there are still more younger people volunteering overall, seniors tend to volunteer more hours and stay with the organization longer than younger adults [8]. Older adults are also more likely to stop volunteering in later life than to start, with those who put in more hours and have been with the organization longer being more likely to stay [18,19]. While these trends paint a clear picture of how volunteering has changed through volunteer participation, there is other research on why these may be the trends. The purpose of this study is to examine the relationships among satisfaction with volunteering in a Foster Grandparent Program, volunteer demographics, and quality-of-life of older adults. Through this line of inquiry, we will better understand the aspects of volunteering that directly affect seniors’ quality of life. This research also examines the program’s impact on their sense of well-being and the satisfaction of those volunteering.

### 1.2. Theories Related to Volunteerism

Studies have shown that volunteering in older age is attributed to a more satisfying retirement and better health outcomes. Most of the research in this area has been credited to Activity Theory. Activity Theory states that the more active a senior is, the more satisfaction they have in life, and this positively increases their self-image [8,9]. For those just retiring, keeping up with activity is a way of aiding the adjustment from working to retirement [8,20,21]. Additionally, working and volunteering have shown positive results in the prevention of mental decline in older adults [6,22,23]. In senior surveys, when asked about their state of mind and regard for life, those who volunteered or had a strong organizational tie were more likely to report less depression and felt they had improvement in quality of life [4,5,6]. Therefore, the more invested seniors are in their volunteer programs, the more likely they are to age better, leading to better health benefits and less risk of mortality. Furthermore, studies show there is an increased risk of mortality in seniors who feel of little use to others [6]. In sum, volunteering is shown to be a good way of allowing seniors to be active and feel useful.

The access to new activities and challenges in volunteering can bring a sense of expansion and growth mentally. This is also known as Self-Determination Theory. Self-Determination Theory states that humans tend to do things that will help them grow and challenge them as a person [10]. Most of these studies look at how volunteering affects people’s feelings of values being met and connectedness with their environment. Research on this theory of volunteering is limited. However, there is important research showing that people who feel challenged to volunteer feel internal satisfaction [10,24]. These internal factors also help determine whether a senior will stay with a certain program or if they will move on to something else that satisfies their growth in other ways. A recent study found that the self-determination motivations behind volunteering were values, understanding, and social justice [24]. While Self-Determination Theory may be a large part of the motivation behind volunteering, it is not the only theory.

Alongside the above-mentioned Activity and Self-Determination Theory, many researchers have looked at Socioemotional Selectivity Theory (SST) as an explanation for why people volunteer. Socioemotional Selectivity Theory states that as individuals age, they are driven towards things that bring about personal growth and enjoyment. Consequently, they choose their activities based on their emotional response. When perceived contributions to volunteering are high, it fulfills a need of altruism, and is associated with better mental health [14]. This value attached to altruism also contributes to the volunteer’s sense of satisfaction in life. This is especially true for people of lower socioeconomic status who have been shown to benefit more from volunteering than those of higher status [14,25]. When looking at trends between younger and older volunteers, it was found that while both groups volunteer according to their values of altruism, younger people value their careers more than their value for altruism [20,26]. Some studies show that some elderly people leave volunteering due to other activities having a higher priority [14,26]. However, Wei et al. disagree with the previous studies, claiming seniors do not leave due to higher priorities, but mainly due to deteriorating health [17]. However, it can be argued that connectedness created by the socialization involved in volunteering helps satisfy a social need, which has been attributed to better health outcomes for older adults [27]. Though there are varying views on Socioemotional Selectivity Theory when it comes to seniors volunteering, there is more research to support positive mental and physical well-being for seniors.

Ultimately, most studies have shown volunteering positively affects older adults in both physical and mental health outcomes. They have also demonstrated there is motivation for older adults to participate in formal volunteering, leading to a better quality of life for both volunteers and those who receive the services. It is important to note that not all volunteer experiences are positive, as emotional and financial costs can be incurred. This research will fill in gaps by looking at the FGP in the High Country and examining quality-of-life changes. Thus, this research continues the investigation of previous literature in explaining the motivations behind volunteering and how volunteering can affect seniors.

## 2. Materials and Methods

The design of this study is a secondary data analysis. This research uses secondary data collected through surveys to be used originally for a governmental program grant to be submitted by the FGP, USA. The surveys were given to the volunteers of the FGP organization to gain a better understanding on the effects of the organization on the volunteers. The sample consisted of senior volunteers participating in the FGP who were at least 55 years old or older. Ninety-three volunteers who were actively participating in the FGP were enrolled in the study. The surveys use a five-point Likert scale rating from “it’s much worse” to “it’s much better”, asking them to self-rate their lives since they began participating in the program. Also, the survey asks participants how much they would associate this change with their time participating in the volunteering organization, ranging from “not responsible at all” to “fully responsible”. The survey also focuses on volunteers’ satisfaction with various areas of the program, with responses from totally dissatisfied to totally satisfied. This research works within the framework of these FGP surveys not given by us, and thus we had no control over the demographics collected or questions asked.

For data analysis, this research will focus on using statistical analysis through SPSS version 29. We analyzed these surveys by looking at how quality of life for the seniors correlates with demographics and satisfaction with experience. Through the analysis of this data, we will gain clearer insights as to how different aspects of an FGP volunteer’s life is affected through their work as a volunteer. All examinations were run at a significance level of *p* < 0.05, due to the size of the sample. Data analysis related to the demographic variables were age, gender, and years of service in the FGP. To examine the demographics, we first separated the age groups into four different categories: pre-old (55–64 years old), young-old (65–74 years old), middle-old (75–84 years old) and old-old (85+ years old), to obtain a better understanding of how volunteering effects these different stages of aging. Satisfaction with the program was measured by the FGP assignment, FGP staff, volunteer site supervisors, training, and overall satisfaction. Cross-tabulation and chi-square examined the relationship between demographic data and quality of life. Pearson correlations determined the relationship between satisfaction with the FGP and quality-of-life measures. In addition, a frequency distribution was examined regarding the answer to the question “in general, to what extent do you credit any change to your quality-of-life to your participation in this program?” This informed us as to how much the older adult volunteers attributed the changes in their quality of life to volunteering in the program. Additional details about QOL measurements are found in Appendix A.

## 3. Results

Study data were separated into three categories—demographics, quality of life, and satisfaction with the program (Table 1). This research uses cross-tabulation and correlation to look at the relationship between these three categories. It will focus on how demographics and satisfaction with the program relate to the changes in the quality of life for the volunteers. The three different categories are shown below.

### 3.1. Demographics and Quality of Life

The demographic data used were age, gender, and years of service, and were tabulated by frequencies. Quality of life was measured by looking at the following survey self-reported frequencies for the answer yes to the questions above in the column “Quality of Life”. When using cross-tabulation to determine the relationship between demographics and quality-of-life, it was shown that neither age categories nor years of service were significantly related with any quality-of-life factors. The survey question “feeling someone is looking out for your welfare” revealed a significant relationship in regard to gender. The test showed a chi-square of 0.002. The relationship between gender and “feeling someone is looking out for your welfare” was shown to be significant at *p* < 0.01 level, although this may be unreliable, as the sample size for males was only 3 out of 93 volunteers.

### 3.2. Satisfaction with FGP Assignment and Quality of Life

The test indicated that satisfaction with one’s assignment was correlated with many changes in quality of life. Satisfaction in assignment revealed a weak positive correlation with “feeling you have purpose in life” and “physical health” at a significance of *p* < 0.05, and “feeling you can make a positive difference in another’s life” and “the amount of pleasure you gain from daily activities” at a significance level of *p* < 0.01. It was also shown to have a moderate positive relationship with “sense of well-being” and “overall quality-of-life changes” at a significance of *p* < 0.01.

### 3.3. Satisfaction with FGP Staff and Quality of Life

The second correlation examined the relationship between FGP staff and quality-of-life changes. The results indicated that satisfaction with the staff was correlated with 10 of the quality-of-life factors. There were weak, positive correlations between satisfaction and “sense of accomplishment”, “feeling you have purpose in life”, “feeling you can make a positive difference in another’s life”, “sense of well-being”, “physical health” and “overall quality of life” changes at a significance level of *p* < 0.05, and “your feeling someone is looking out for your welfare” and “your sense of self-esteem” at a significance level of *p* < 0.01. Satisfaction was also shown to have a medium positive relationship with “looking forward to each new day” and “the amount of pleasure gained from daily activities” at a significance level of *p* < 0.01.

### 3.4. Satisfaction with Volunteer Site Supervisors and Quality of Life

Satisfaction with volunteer site supervisors only revealed four correlations with the quality-of-life changes in the volunteers. It showed that satisfaction with volunteer supervisors had a weak positive relationship with “feeling you can make a difference in another’s life”, “feeling someone is looking out for your welfare”, and “your physical health”, at a significance of *p* < 0.05, and sense of well-being at a significance of *p* < 0.01. Finally, satisfaction with volunteer site supervisors did not correlate with “overall changes in quality of life” given a significance level of *p* = 0.128.

### 3.5. Satisfaction with Training and Quality of Life

Pearson’s correlations indicated that satisfaction with training was significantly correlated with eight factors related to quality of life. There was a weak positive relationship with “sense of well-being”, “your feeling someone is looking out for your welfare”, “your physical health” and “overall quality-of-life changes” at a significance level of *p* < 0.05, and “your feeling you can make a difference in another’s life” at a significance level of *p* < 0.01. Satisfaction was also shown to have a moderate positive relationship with the “amount of pleasure you gain from daily activities”, “sense of self-esteem” and “ability to make ends meet” at a significance level of *p* < 0.01

### 3.6. Satisfaction with Overall Experience and Quality of Life

Results indicated statistically significant correlations between satisfaction with the overall experience and 10 of the factors of quality of life. Weak, positive relationships were identified with volunteers’ “sense of accomplishment” at a significance level of *p* < 0.05 and “looking forward to each new day” at a significance level of *p* < 0.01. It was also shown to have a relationship with “feeling you have purpose in life”, “feeling you can make a difference in another’s life”, “the amount of pleasure gained from daily activities”, “sense of self-esteem”, “ability to make ends meet”, “sense of well-being”, “your feeling someone is looking out for your welfare” and “overall quality-of-life changes”, with a moderate positive relationship at a significance level of *p* < 0.01.

### 3.7. FGP Participation and Perceived Change in Quality of Life

To consider the influence of FGP participation on participants’ quality of life, frequency distributions were examined. Frequencies indicated that 40.9% said that the FGP was fully responsible for their changes in quality of life, while 44.1% said that the program was almost fully responsible for change in their quality of life. These findings reveal that the majority of FGP volunteers attributed changes in the quality of life to participation in the FGP program. Frequencies showed that 44.1% said that the program was almost fully responsible, while another 40.9% said the program was fully responsible for their changes in quality of life. These findings reveal that most of the volunteers in the program say it is responsible for the changes in quality of life since they joined the FGP program.

## 4. Discussion

The results are consistent with the previous literature on Activity Theory, Self-Determination Theory, and Socioemotional Selectivity Theory. This research demonstrates a correlation with satisfaction in volunteering and volunteers’ perception of their changes in quality of life. The volunteer impact scales evaluate this relationship within the Foster Grandparents Program (FGP) of the High Country and examine factors that appear to lead to these changes. The research found a strong relationship between volunteers’ experience with the program and an increase in quality of life, while showing that demographics had a weak relationship with quality-of-life changes.

### 4.1. Demographics and Quality of Life

This research illustrated that demographics did not correlate well with any factors of quality-of-life changes. This is unsurprising as many studies have come to the same conclusion that quality-of-life changes occur at the same rate, no matter the demographics of the volunteer [16,17]. This also brings to attention the gender disparities within the FGP, where only 3 volunteers out of 93 were men. This could be due to volunteering being strongly associated with altruism, which is more associated with women than men [28]. Another factor could be Socioemotional Selectivity Theory, in that women have a higher priority for altruism while men hold higher priorities in other activities, such as careers [20,26]. While years of service did not correlate with any quality-of-life factors, it has been demonstrated that the changes in quality of life can increase the longevity of the volunteers. In future research, the inclusion of additional demographic variables of volunteers, such as socioeconomic status and level of education, may help gain a better understanding of the relationship between volunteering, social class, and changes in quality of life.

### 4.2. Satisfaction with Assignments and Quality of Life

Satisfaction with program assignments illustrates a connection with various quality-of-life changes. The association between a sense of purpose in life, the amount of pleasure you gain from daily activities, physical health, and sense of well-being, all show that satisfaction in assignment can be a determinate for better mental health and lower symptoms of depression. When you are satisfied with where you are assigned, this creates less stress, which leads to a more positive experience for volunteers [14]. Being able to feel like you can make a positive difference in another’s life shows that volunteer assignments can create a feeling of altruism. Satisfaction with volunteer assignments were shown to be highly significant with overall changes in quality of life.

### 4.3. Satisfaction with FGP Staff and Quality of Life

According to the correlation test, satisfaction in FGP staff is directly related to ten of the eleven quality-of-life factors. Some of the most significant factors are feeling someone is looking out for your welfare, your sense of self-esteem, physical health, feelings of accomplishment, and feeling you have purpose in life. When satisfied with FGP staffing, an individual’s sense of identity and community increased. When satisfaction with FGP staff is high, it can positively affect how individuals feel about themselves and the work they are doing. The “feeling that someone is looking out for your welfare” shows the volunteers’ sense of worth to the staff. This is important, as isolation and loneliness are common among seniors, and feeling someone cares about them can reduce these actions and feelings. It has been shown that organizational support can have a positive impact on socioemotional and health benefits for volunteers [14]. Research correlates isolation and feelings of uselessness with higher rates of depression in seniors, and having a community you feel a part of can counteract these emotions [29,30]. The FGP staff provide training to the volunteer seniors once a month, teaching different skills, thus giving the seniors a sense of accomplishment and purpose in life. By positively influencing a volunteer’s overall changes in quality of life, this confirms just how important satisfaction with the FGP staff is in influencing the feeling of belonging to the senior volunteer community. This sense of community created by the FGP staff carries over into the volunteer training they receive.

### 4.4. Satisfaction with Training and Quality of Life

Experience with training received again exposes a relationship of satisfaction to sense of belonging. These four-hour sessions are taught by the FGP staff, and teach volunteers life skills alongside skills for their assignments. This provides the volunteer with a mentor to learn from and with whom they grow their skills as a volunteer. This is also likely to be the reason this satisfaction correlated with a volunteer’s feeling they can make a positive difference in another’s life. These training sessions offer educational information for the volunteers, challenging them and creating growth. According to Self-Determination Theory, this challenge and growth are likely to retain volunteers. These training session help create a motivation in volunteers of values for volunteering, understanding through education, and social justice through action in the community, all of which are important for the feeling of personal growth [24]. Other studies have also shown that volunteer training can be a strong factor in retaining volunteers [14]. When volunteers feel they can succeed, this increases volunteers’ sense of self-esteem and allows them to take away more enjoyment from their activities, thus showing satisfaction with received training. While the above satisfactions have been shown to have high correlation rates, one of the low-ranking correlations is satisfaction with volunteer site supervisors.

### 4.5. Satisfaction with Volunteer Site Supervisors and Quality of Life

Satisfaction with volunteer site supervisors was correlated with only four of the quality-of-life factors. Many of these factors have been referred to above, and conclude that site supervisors may play a role in community, their sense of altruism, and mental and physical health of the volunteers. While supervisors play a minimum role compared to the FGP staff, the data display many of the same characteristics when it comes to changes in quality of life. Due to these similarities and only correlating with four factors, this research has concluded that satisfaction with the volunteer site supervisor is not a good indicator for changes in quality of life, and does not correlate with overall changes in quality of life for the volunteers.

### 4.6. Satisfaction with Overall Experience and Quality of Life

The volunteers’ overall experience satisfaction was shown to correlate with all quality-of-life factors except for physical health. Overall satisfaction with the experience of the volunteers showed improvement in depression symptoms, a sense of belonging with the program, their feeling of being supported, and that they are given the tools necessary to succeed. Thus, overall satisfaction with the volunteer experience was shown to be the best indicator of changes in quality of life. This is supported by research that shows volunteer programs can be changed to maximize the benefits to the volunteers [9]. This suggests satisfaction with the program increases overall changes in quality of life and suggests overall experience is vitally important in predicting outcomes of volunteers.

### 4.7. Limitations

The study’s potential for generalizability is a significant limitation, due to the small sample size. Moreover, this research focuses only on one FGP program in the U.S., and thus it used a single site of data collection. In addition, this study did not include a control group to compare the pre–post effects of volunteers’ experiences. More in-depth research should be conducted, as there are limitations to this research due to the survey used and its use of limited demographics.

## 5. Conclusions

In conclusion, this research has substantiated the idea that overall satisfaction with the program influences volunteers’ quality of life. Volunteering is often difficult and time-intensive, and people volunteer for a variety of different reasons. Studies have shown that motivation to volunteer is closely tied to one’s values, such as moral, religious, and personal beliefs [9]. Others have shown us that older adults volunteer to gain skills to advance their career [19]. However, other scholars have also found that older adults volunteer to better one’s mental and physical health [6]. This research supports previous studies showing volunteering as having a positive impact on the seniors who volunteer. We add to these studies by showing that by providing them with FPG supervisors who are supportive of their efforts to volunteer with FGP in the first place leads to better quality-of-life outcomes. The results of this study supports the relationship between satisfaction with the volunteer experience and the factors associated with quality of life.

The FGP program, as it represents older adult volunteers, serves as an excellent population for understanding the relationship between aging and the current issues facing an aging population. Aging successfully requires that older adults maintain their mental and physical well-being. Participating in an FGP program provides volunteers with several benefits, including a sense of purpose and fulfillment by mentoring children, improved mental and physical well-being through social interaction and activity, a small stipend, access to training, reduced feelings of isolation, and the opportunity to contribute positively to their community by supporting children with special needs [2]. Maintaining funding for this program should be a priority for the United States, because it can lead to positive quality-of-life outcomes for older adult volunteers.

## Figures and Tables

**Table 1 healthcare-13-00230-t001:** Data Categories.

Demographics	Quality of Life	Satisfaction with Volunteer Experience
1. Age categories 2. Gender 3. Years of service	1. Feeling of Accomplishment 2. Feeling of purpose in life 3. Physical Health 4. Sense of well-being 5. Pleasure in performing everyday activities 6. Sense of self-esteem 7. Feeling you can make a positive difference in another’s life 8. Feeling that someone is looking out for your welfare 9. Looking forward to each new day 10. Ability to make ends meet 11. Overall quality of life	1. FGP Assignment 2. FGP Staff 3. Volunteer site supervisors 4. Training received 5. Overall experience

## Data Availability

The data that support the findings of this study are available from the corresponding author, [A.S.], upon reasonable request.

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
