# Peer review of "Foster Grandparent Programs’ Impact on the Quality-of-Life of Older Adult Volunteers"

_healthcare, 2025, doi:10.3390/healthcare13030230_

Round 1

Reviewer 1 Report

Comments and Suggestions for Authors

Overall, this is a nicely written manuscript that reports on the perceived quality of life benefits among older adults volunteering with the Foster Grand Parents program. The research question is clearly stated and the manuscript is well organized and provides sufficient background on volunteering during retirement. At its core, this study represents a program evaluation and is a strong example of how applied research can be used to assess programs while also contributing to larger theoretical discussions.

I'm curious as to why the authors did not construct a scale(s) to measure the different dimensions of QOL that were measured in the manuscript. Due to the applied nature of the research, advanced statistical analysis may not be warranted. 

The tables, as presented are somewhat hard to follow. I'm not sure if that's due to the journals style guidelines or not. I recommend: 1) Including a Sample Characteristics table that includes demographics and other relevant information on the same 2) combining individuals correlation/crosstab tables into larger tables. This would help organize the flow of the findings section.

As I've said, the paper is well written. I didn't find any typos or major grammatical errors. However, I did notice the use of the word "I" in the methods section. As this paper is co-authored, I was unsure of who that referred to.

Based on my reading of the manuscript, I believe that it is suitable for publication in its current form but would benefit from the suggestions above. 

Please let me know if you have any additional questions.

Author Response

1. Summary

Thank you very much for taking the time to review this manuscript. Please find the detailed responses below and the corresponding revisions/corrections in the re-submitted file.

2. Questions for General Evaluation

Reviewer’s Evaluation

Response and Revisions

Does the introduction provide sufficient background and include all relevant references?

Yes

Are all the cited references relevant to the research?

Yes

Is the research design appropriate?

Can be improved

The research design was edited to make clearer the steps we took during the research process. This should address any questions about the appropriateness of the method and analysis used.

Are the methods adequately described?

Yes

Are the results clearly presented?

Can be improved

The results and conclusion section was added to and clarified by listing the findings in sequential order and making sure to address QOL in the last couple of paragraphs of the paper.

Are the conclusions supported by the results?

Yes

3. Point-by-point response to Comments and Suggestions for Authors

Comments 1: Overall, this is a nicely written manuscript that reports on the perceived quality of life benefits among older adults volunteering with the Foster Grand Parents program. The research question is clearly stated and the manuscript is well organized and provides sufficient background on volunteering during retirement. At its core, this study represents a program evaluation and is a strong example of how applied research can be used to assess programs while also contributing to larger theoretical discussions.

I'm curious as to why the authors did not construct a scale(s) to measure the different dimensions of QOL that were measured in the manuscript. Due to the applied nature of the research, advanced statistical analysis may not be warranted.

Response 1: Thank you for your supportive and valuable feedback on this manuscript. Also, thank you for pointing out the question about not using a QOL scale. Unfortunately, we cannot separate this from the survey data collected by the program supervisors.

Comments 2: The tables, as presented are somewhat hard to follow. I'm not sure if that's due to the journals style guidelines or not. I recommend: 1) Including a Sample Characteristics table that includes demographics and other relevant information on the same 2) combining individuals correlation/crosstab tables into larger tables. This would help organize the flow of the findings section.

Response 2: We have removed the tables per your comments and the journal editors' comments about the readability and flow of data in the article. Moreover, we clarified the demographic data of the participants in the paper as well.  

4. Response to Comments on the Quality of English Language

Point 1: As I've said, the paper is well written. I didn't find any typos or major grammatical errors. However, I did notice the use of the word "I" in the methods section. As this paper is co-authored, I was unsure of who that referred to.

Response 1: We have removed all references to the word “I” in the methods section. Thank you for pointing this out; it is much appreciated!

Reviewer 2 Report

Comments and Suggestions for Authors

Thanks to the authors for a very good topic and results.

However, the paper is very incomplete in terms of completeness and writing.

1. I don't know what you're trying to say in the theoretical background in the introduction - a lot of different information and too many paragraphs. 

2. the research methodology is highly inappropriate

2-1. No description of study subjects

2-2. No description of the tools used in the study.

In the research method, there is only a brief description of the research instrument, data collection, and analysis.

3. I think it would be easier to understand if the table of results showed which paragraph it was talking about.

There are too many tables and results, it's confusing.

3. There are also items that will change the number of people in your results.

There are two reasons for this: either they didn't take the survey, or we deleted them because something was statistically wrong.

To eliminate these questions, you need normalization.

4. the preceding narrative is very weak in comparison to the discussion narrative.

You need to add a more scientific, evidence-based narrative about why we got the results we did.

Author Response

Response to Reviewer 2 Comments

1. Summary

Thank you very much for taking the time to review this manuscript. Please find the detailed responses below and the corresponding revisions/corrections in the re-submitted file.

2. Questions for General Evaluation

Reviewer’s Evaluation

Response and Revisions

Does the introduction provide sufficient background and include all relevant references?

Yes

Are all the cited references relevant to the research?

Yes

Is the research design appropriate?

Must be improved

The research design was edited to clarify the steps we took during the research process. This should address any questions about the appropriateness of the method and analysis used. Moreover, survey data has been used in prior studies to examine QOL of volunteers. See “The effect of volunteer-led activities on the quality of life of volunteers, residents, and employees of a long-term care institution: a cohort study” by Dandes-Guimaraes et al. (2023) and their use of the same research design and method used.

Are the methods adequately described?

Must be improved

In the methods section, we added additional clarifying data about participants (sample) and methodology.

Are the results clearly presented?

Can be improved

The results and conclusion section was added to and clarified by listing the findings in sequential order and making sure to address QOL in the last couple of paragraphs of the paper.

Are the conclusions supported by the results?

Must be improved.

Once again, we added additional information in the conclusion to clarify our findings.

3. Point-by-point response to Comments and Suggestions for Authors

Comments 1: I don't know what you're trying to say in the theoretical background in the introduction - a lot of different information and too many paragraphs.

Response 1: Thank you for your valuable feedback on this manuscript. The theoretical background in the introduction is a typical literature review, where we explore what theories have been used to explain older adult volunteers and the impacts upon QOL of the volunteers.

Comments 2: the research methodology is highly inappropriate

2-1. No description of study subjects

2-2. No description of the tools used in the study

Response 2: We have clarified the demographic data of the participants in the paper in the introduction of the paper and in further detail in the sample/methods section.

2-1 “See lines 39-48: “The Foster Grandparents Program (FGP) is one of the programs run by the Appalachian Senior Programs. The FGP allows older adults to provide support services to youth who have special emotional, social, or educational needs. Older adult volunteers in these programs devote 20 to 40 hours of their week to help these kids. The Appalachian Senior Programs is federally funded by the Senior Corps program and allows older adults within the High Country consisting of Allegheny, Ashe, Avery, Watauga and Wilkes counties of North Carolina the opportunity to volunteer time to help in their community. This community-based program impacts about 100 older adult volunteers and helps about 428 children [7]. Volunteering through this program is important as it betters the lives of the volunteers and the “at-risk” children they assist.”

Also, please see lines 177-181 “Data analysis related to the demographic variables were age, gender, and years of service in the FGP. To examine the demographics, we first separated the age groups into four different categories: pre-old (55-64 years old), young-old (65-74 years old), middle-old (75-84 years old) and old-old (85+ years old) to get a better understanding of how volunteering effects these different stages of aging.”

And, lines 167-162 “     This research uses secondary data collected by the FGP. The data collected from these surveys were originally used for a governmental program grant for FGP. Surveys were given to the volunteers of the FGP organization to gain a better understanding on the ef-fects of the organization on the volunteers. The sample consisted of senior volunteers par-ticipating in the FGP who were at least 55 years or older. Ninety-three volunteers who were actively participating in the FGP were enrolled in the study.”

2-2 . Please see lines 172-174 “  For data analysis, this research will focus on using statistical analysis through SPSS. We analyzed these surveys by looking at how quality of life for the seniors correlates to demographics and satisfaction with experience.”

Comments 3: I think it would be easier to understand if the table of results showed which paragraph it was talking about.

There are too many tables and results, it's confusing.

Response 3: Thank you for these comments. Based on your feedback and that of the journal editor, we removed most of the tables and presented the data in narrative form.

Comment 4: There are also items that will change the number of people in your results.

There are two reasons for this: either they didn't take the survey, or we deleted them because something was statistically wrong.

To eliminate these questions, you need normalization.

Response 4: The N=93 remained the same throughout the paper, thus x remained a continuous feature for data standardization.

Comment 5:   the preceding narrative is very weak in comparison to the discussion narrative.

You need to add a more scientific, evidence-based narrative about why we got the results we did.

Response 5: Thank you for your feedback. In turn, we added additional information about QOL to the conclusion about why we arrived at the findings we did in that particular section of the paper. We state, “Data shows that adequate training and ongoing support from program staff is essential for volunteers to feel confident in their role, thus improving their QOL.” in lines 378-380.

Reviewer 3 Report

Comments and Suggestions for Authors

my comments are attached. 

Comments on the Quality of English Language

Very good except for 1-2 sentences. 

Author Response

I think the article deals with an important research question from an unstudied population. However, there are a few issues that caught my attention. Addressing these points can strengthen the article.

Thank you for your valuable feedback on our article. We greatly appreciate the time and timeliness of the reviewer. You can find our comments to you italicized in the comments below.

  1. You mentioned in page 1 line 29-36 the benefits of volunteering for seniors. You also mentioned the same line of argument in page 2 lines 89-98. You could mention initially the advantages of prosocial behavior in general which includes donation There may be other articles one can suggest, here is one article that can help you in that regard:
    • Ren, , & Ye, M. (2017). Donations make people happier: Evidence from the Wenchuan earthquake. Social Indicators Research, 132(1), 517-536.
      • Because we focus on volunteering and not the larger topic of prosocial behavior (specifically, donations), we have decided not to include this suggestion. We will, however, include it in future works when looking at the narratives of the volunteers and not just the survey data collected from the FGP supervisors.
  1. You mentioned in detail the advantages of volunteering. However, volunteers often help people who are in adverse circumstances. Depending on the severity of those circumstances, there is evidence that volunteers might be exposed to secondary trauma. You may want to incorporate that aspect as well in the introduction. Here is one article that can point the importance of coping for volunteers and non-profit workers in general.
    • Uǧur, B. (2024). The role of coping strategies in subjective wellbeing indicators of non-profit workers. Voluntary Sector Review, 1-23.
      • This is a significant point, especially when working with troubled children in the K-12 school system. We cannot, however, get to this question with the data that we have to analyze. Therefore, we will use this in future work when looking at the narratives of the volunteers. We do find that they do “talk” about this in their interviews, but not the survey data.
  1. It will be nice to see more details about “the North 143 Carolina High Country program or the Foster Grandparents Program (FGP)”, who are the beneficiaries of the program? What kind of help do they receive in more detail?
    • We have added in the following to expand upon the FGP. Volunteers on line 49 and it states the following:
    • The FGP engages people 55 and older in volunteer service, which can help them stay active and enrich their lives. Volunteers receive benefits such as a tax-exempt hourly stipend, mileage reimbursements, and paid vacation and sick leave. They also receive accident and personal liability insurance, as well as a small death benefit.

Children

  • The FGP provides one-on-one support to children with special needs, which can help improve their academic, social, or emotional development. Volunteers can help children learn to read, tutor them, mentor teenagers, care for premature infants or children with disabilities, and help children who have been abused or neglected.
  1. “Data Collection” part can be made bold or separated clearly from the previous paragraph. The same goes for “Data Analysis” heading.
    • The journal section editor has made this formatting change.
  2. “Also, using the five-point Likert Scale, the survey asks participants how much they would associate this change with their time participating in the volunteering organization, ranging from “not responsible at all” to “fully responsible.”” This sentence is not clear to me. What do you mean by responsibility with regard to participating in volunteering organization?
    • We state, “The survey asks participants how much they would associate this change with their time participating in the volunteering organization, ranging from “not responsible at all” to “fully responsible.” The question is asking them if they believe they are responsible for that change.
  3. “This study aims to examine how participation in the Foster Grandparents Program impacts changes in the quality of life of its volunteers”. This sentence does not belong to results and mentioned already in the methods part.
    • Removed/deleted.
  4. “This research uses cross-tabulation and correlation to look at the relationship between these three categories.” This sentence belongs to the methodology section I guess.
    • Because we talk about the cross-tabulations in this section, we restated it.
  5. Table 2 is not useful in its current You better give the percentages.
    • We have removed most tables and put those in narrative form, per the journal editors' comments.
  6. “It was also shown to have a medium positive relationship with “sense of well- being” and 216 “your overall quality of life changes” at a significance of p<.01.” Why do not you mention the correlation coeficient here? In Table 2, you give

positive relationship.

  • Added in that the correlation coefficient was a weak positive relationship in line 216.
  1. How overall quality of life is measured? How it is worded? With which options, the data is collected is a bit of a puzzle.

The same goes for other quality of life measures. You can give their details in appendix and mention it inside the text that full wording and options are available somewhere.

  • Quality of life is a slippery slope concept, to begin with and has a multitude of debates surrounding sociology and health science literature. Therefore, we measure QOL is explained in lines 180-198 of the manuscript. We state, “We analyzed these surveys by looking at how quality of life for the seniors correlates to demographics and satisfaction with experience.”
  1. The tables look ugly and very raw You could put them in a better shape.
    • We have removed most tables and put these in narrative form.
  2. My final suggestion is to acknowledge your limitations a little
    • We have added in the following to line 384, “The study’s generalizability is a significant limitation due to the small sample size. Moreover, this research focuses only on one FGP program in the U.S., thus it was a single site of data collection. In addition, this study did not include a control group to compare the pre-post effects of volunteers’ experiences. More in-depth research should be conduct-ed as there are limitations to this research due to the survey used and its use of limited demographics.”

Round 2

Reviewer 2 Report

Comments and Suggestions for Authors

1. There's a change in the introduction, but it doesn't yet reveal a clear purpose.

2-2. 2-2 is a statistical analysis method, not a description of the tools used in the study.

3. no change from the previous result.

4. 5's discussion is insufficient.

Author Response

Comment 1: There's a change in the introduction, but it doesn't yet reveal a clear purpose.

Response 1: 

On lines 76-81

"Using volunteer impact scales, this paper will look at the influence of volunteering for the FGP on senior volunteers. Through this, one will better understand the aspects of volunteering that directly affect senior’s quality of life. This research will also examine the program’s impact on their sense of well-being and the satisfaction of those volunteering. Consequently, this paper will show how the FGP impacts seniors and deserves further support."

Also, on lines 36-38, we state, "Countless studies have shown the impact of volunteering on the older adult population, but there are still more questions to ask." in the first paragraph. 

We added, "The purpose of this study is to look at the FGP to see if volunteering has an impact on older adult volunteers' quality of life and sense of well-being," on line 38. 

Comment 2: 2-2 is a statistical analysis method, not a description of the tools used in the study.

Response 2:

Line 201

We do not see any mention of "tools used in the study." We have stated the statistical analyst method on line 201, stating, "For data analysis, this research will focus on using statistical analysis through SPSS." 

At this time, we do not see where this change needs to occur, as we did mention it is a statistical analysis and never mention "tools used." 

Comment 3: 

3. no change from the previous result. 

Prior comment is below: 

I think it would be easier to understand if the table of results showed which paragraph it was talking about.

There are too many tables and results, it's confusing.

Response 3: Thank you for these comments. Based on your feedback and that of the journal editor, we removed most of the tables and presented the data in narrative form.

Comment 4: 

5's discussion is insufficient.

Response 5: 

Three is no 5 in our manuscript. We end with four (4). Moreover, we need additional information as to what is needed and why it is insufficient. 

Thank you, once again, for your valuable feedback on our manuscript. It is very much appreciated. 

Reviewer 3 Report

Comments and Suggestions for Authors

My comments are attached. 

Author Response

Comment 1: 

Reviewer's Comment from Round 1: You mentioned on page 1, lines 29-36, the benefits of volunteering for seniors. You also mentioned the same line of argument in page 2 lines 89-98. You could mention initially the advantages of prosocial behavior in general which includes donation There may be other articles one can suggest, here is one article that can help you in that regard:

Ren, , & Ye, M. (2017). Donations make people happier: Evidence from the Wenchuan earthquake. Social Indicators Research, 132(1), 517-536.

Our Response to Review Round 1: Because we focus on volunteering and not the larger topic of prosocial behavior (specifically, donations), we have decided not to include this suggestion. We will, however, include it in future works when looking at the narratives of the volunteers and not just the survey data collected from the FGP supervisors.

Reply from Reviewer Round 2: I understand your point. However, when I read that part again. I see that first you talk about the Senior Corps, then you move to the benefits of volunteering, then you talk about the foster grandparents program. Moreover, this benefits of volunteering has been mentioned in other parts of the article as I mentioned in the previous round of revisions. The current structure of the article has issues with regards to flow.

Our Response 1 to Review Round 2: 

We discuss the benefits of volunteering with Senior Corps and the overall benefits of older adult volunteers. We move from specific (the program we look at) to general (overall impact of volunteering to seniors research). 

We did add on lines 40-41, "The Foster Grandparents Program (FGP) is one of the programs run by the Appalachian Senior Programs and funded by Senior Corps," to make it clearer that we are talking about the program we study in this research and not volunteering in general. 

Comment 2: 

Reviewer Round 1 Comment: You mentioned in detail the advantages of volunteering. However, volunteers often help people who are in adverse circumstances. Depending on the severity of those circumstances, there is evidence that volunteers might be exposed to secondary trauma. You may want to incorporate that aspect as well in the introduction. Here is one article that can point the importance of coping for volunteers and non-profit workers in general.

Uǧur, B. (2024). The role of coping strategies in subjective wellbeing indicators of non-profit workers. Voluntary Sector Review, 1-23.

Our Reply to Round 1 Reviewer: This is a significant point, especially when working with troubled children in the K-12 school system. We cannot, however, get to this question with the data that we have to analyze. Therefore, we will use this in future work when looking at the narratives of the volunteers. We do find that they do “talk” about this in their interviews, but not the survey data.

Reply from Reviewer Round 2: To be honest, when I read your article, it suggests that volunteering is always a good thing. But, I am not convinced that it is always a good thing for everyone. Moreover, if that is the case, why should you research the impact of Foster Grandparents program, you could easily predict it to be a positive thing.

Our Response 2 to Reviewer Round 2: 

We agree and added the following sentence to highlight the point that not all volunteering has a positive impact on volunteers. See lines 166-167, where we state, "It is important to note that not all volunteer experiences are positive, as emotional and financial costs can be incurred." 

Comment 3: 

Reviewer Round 1 Comment: “This research uses cross-tabulation and correlation to look at the relationship between these three categories.” This sentence belongs to the methodology section I guess.

Our Response from Round 1: Because we talk about the cross-tabulations in this section, we restated it.

Reply from Reviewer Round 2 Review: This cross-tabulation and correlation are mentioned three times in the article, one in the abstract, one in the methodology, and one in the results section, almost in the same way. You can come up with a clever way of explaining your cross-tabulation results without writing the same thing again and again.

Our Response 3 to Round 2 Review: To maintain consistency throughout the manuscript, we use cross-tabulation in the abstract, methods section, and results section. 

Comment 4: 

Reviewer comments from Round 1:
How overall quality of life is measured? How it is worded? With which options, the data is collected is a bit of a puzzle.
The same goes for other quality of life measures. You can give their details in appendix and mention it inside the text that full wording and options are available somewhere.

Our Response from Round 1: Sociology and health science literature. Therefore, QOL is explained in lines 180-198 of the manuscript. We state, “We analyzed these surveys by looking at how the quality of life for the seniors correlates to demographics and satisfaction with the experience.”

Reviewer Reply from Round 2: Your response is not adequate. Your outcome variable is QOL, therefore, you can not treat it like a background variable. it is important that the reader can check how these variables are worded. I mean you need to give details of the constructs in Table 1.

Our Response 4 to Round 2 Review: We have added the following to tables 1-5 in our supplementary tables file that explain how we measured our outcome variable QOL. We also added the following statement in the manuscript, "Additional details about QOL measurements are found in Tables 1-5 in the supplemental materials." to lines 220-221 in the manuscript. 

Table 1 - **We measure quality of life by looking at the response to questions about “purpose, positive difference, and pleasure” from the volunteering experiences with “Satisfaction of the FGP assignment.”

Table 2 - **We measure quality of life by looking at the response to questions about “purpose, positive difference, and pleasure” from the volunteering experiences with “Satisfaction of the FGP assignment.”

Table 3 - **We measure quality of life by looking at the response to questions about “positive difference, sense of well-being, and changes in overall quality of life (self-reported) ” from the volunteering experiences with “Satisfaction with volunteer site supervisor.”

Table 4 - **We measure the quality of life by looking at the response to questions about “positive difference, pleasure, physical health impacts, and overall quality of life (self-reported)” from the volunteering experiences to “Satisfaction with training received.”

Table 5 - **We measure quality of life by looking at the response to questions about “sense of accomplishment, purpose, positive difference, making ends meet, and sense of well-being” from the volunteering experiences to the overall volunteering experience.